# CFMAE: A COARSE-TO-FINE VISION PRE-TRAINING FRAMEWORK FOR HIERARCHICAL REPRESENTATION LEARNING

## ABSTRACT

Prevailing self-supervised learning paradigms, such as contrastive learning (CL) and masked image modeling (MIM), exhibit opposing limitations. CL excels at learning global semantic representations but sacrifices fine-grained detail, while MIM preserves local details but struggles with high-level semantics due to its semantically-agnostic masking, leading to "attention drift". To unify the strengths of both, we propose CFMAE, a coarse-to-fine vision pre-training framework that explicitly learns a Masked AutoEncoder over a hierarchy of visual granularities. CFMAE synergistically integrates three data granularities: semantic masks (coarse), instance masks (intermediate), and RGB images (fine). We enforce the coarse-to-fine principle through two key innovations: (1) a **cascaded decoder** that sequentially predicts scene-level semantics, then object-level instances, and finally pixel-level details, ensuring a structured feature refinement process; and (2) a **progressive masking** strategy that creates a dynamic training curriculum, shifting the model's focus from coarse scene context to fine local details. To support this, we construct a large-scale, multi-granular dataset by generating high-quality pseudo-labels for ImageNet-1K. Extensive experiments show that CFMAE achieves state-of-the-art performance on image classification, object detection, and semantic segmentation, validating the effectiveness of our hierarchical design in learning more robust and generalizable representations.

## 1 INTRODUCTION

Among the diverse paradigms in computer vision pre-training, contrastive-based (He et al., 2020; Chen et al., 2020; Caron et al., 2021) and reconstruction-based (Bao et al., 2021; He et al., 2022; Xie et al., 2022b) self-supervised learning have been particularly influential. Despite their tremendous success, both paradigms exhibit inherent, almost opposing, limitations that curtail their ability to learn truly comprehensive and universal visual representations.

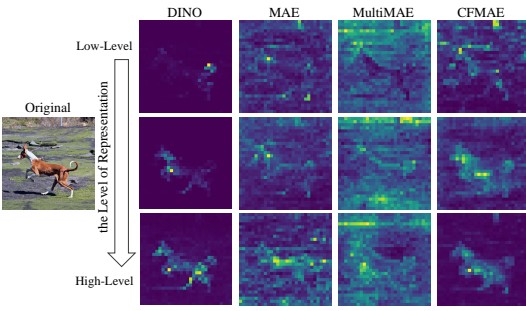

Figure 1: Attention maps from different methods, highlighting their representational focus. DINO excels at capturing high-level semantics, while MAE and MultiMAE's attention is directed toward low-level features. In contrast, our CFMAE effectively captures features across all levels, successfully building a more robust hierarchical representation.

Contrastive learning (CL), which pulls together global features from different views of an image, excels at learning high-level semantic representations. This makes it highly effective for image-level tasks like classification and produces clean, object-centric attention maps as shown for DINO (Caron et al., 2021) in Figure 1. However, this strong focus on high-level semantics inherently limits the capture of fine-grained spatial information. The loss of low-level detail can limit its performance on dense prediction tasks that require precise localization and texture understanding, such as object detection and semantic segmentation (Zhang et al., 2020; Mahajan et al., 2018).

In contrast, Masked Image Modeling (MIM) has emerged to learn rich spatial information by reconstructing masked patches. While MIM's pixel-level objective preserves fine-grained details, its semantically-agnostic random masking strategy struggles to guide the model toward semantically critical regions (Kakogeorgiou et al., 2022; Li et al., 2022a). The model often allocates significant representational capacity to reconstructing simple, low-level areas, while only crudely modeling the core objects of interest. As shown in Figure 1, MAE (He et al., 2022) and MultiMAE (Bachmann et al., 2022) produce diffuse attention maps that fail to focus on salient objects.

We refer to these phenomenon as "attention drift", where pre-training methods develop a biased focus on certain representational levels, thus failing to learn a complete, hierarchical understanding of the visual world. An intuitive solution is to introduce semantic guidance to focus the reconstruction on salient foreground objects, a strategy shown to improve downstream performance (Li et al., 2022a; Sick et al., 2025). However, existing methods often rely on inaccurate, self-generated attention maps for this guidance. While modern segmentation models (Kirillov et al., 2023; Ren et al., 2024) can provide far more precise masks, we argue that simply using them to distinguish foreground from background offers only a rudimentary form of semantic guidance. A truly comprehensive visual understanding is inherently multi-granular, requiring a model to perceive the world at multiple levels of abstraction simultaneously—from coarse scene layouts to intermediate object instances and fine-grained pixel details. This hierarchical, coarse-to-fine principle is not only a long-standing and effective strategy in computer vision (Lin et al., 2017; Jiang et al., 2022) but is also deeply rooted in the efficient processing pipeline of biological vision (Navon, 1977; Serre, 2014), offering proven advantages in learning speed and generalization (Cho et al., 2021; Chen et al., 2023b).

An ideal pre-training framework should therefore unify the high-level semantic understanding of CL with the fine-grained detail preservation of MIM through explicit hierarchical guidance. To this end, we propose a coarse-to-fine strategy across three granularities: beginning with coarse, scene-level semantics to establish spatial context; then using intermediate, object-level guidance to focus on key regions; and finally, returning to fine, pixel-level reconstruction to capture local details and enhance the overall representation. Based on this strategy, we propose CFMAE, a vision pre-training framework that deeply integrates the coarse-to-fine principle. To provide the model with hierarchical guidance signals, we incorporate three visual modalities of different granularities: RGB images (pixel-level), instance segmentation masks (object-level), and semantic segmentation masks (scene-level). And we enforce the coarse-to-fine principle through two synergistic innovations.

First, we design a cascaded decoder, as opposed to a traditional parallel structure (Bachmann et al., 2022). It first predicts scene-level semantic masks, then object-level instance masks, and finally reconstructs pixel-level RGB images. This pipeline ensures that the feature refinement process strictly follows a path from high-level abstractions to low-level details. To combat the "attention drift" caused by random masking, we design a dynamic masking strategy that aligns with the objectives of our cascaded decoder. During training, the focus of the masking follows a carefully designed curriculum, smoothly transitioning from semantic guidance (focusing on scene regions) to instance guidance (focusing on objects), and finally refining with random masking (focusing on local details).

To support our framework, we construct a large-scale multi-granular dataset by generating high-quality aligned instance and semantic segmentation pseudo-labels for all 1.28M images in ImageNet-1K. Through the deep synergy of the cascaded decoder and progressive masking, CFMAE embeds the coarse-to-fine principle into every stage of pre-training. This effectively overcomes the limitations of previous paradigms, enabling the model to learn more robust and generalizable hierarchical visual representations. As visually evidenced in Figure 1, our method produces attention maps that perform well across different representation levels, thereby resolving the "attention drift" issue and validating the successful construction of a true hierarchical visual representation. Extensive experiments show CFMAE achieves significant performance gains across multiple vision tasks, such as image classification, object detection, and semantic segmentation.

## 2  RELATED WORKS

### 2.1  MASKED IMAGE MODELING

Masked Image Modeling (MIM) has become a dominant self-supervised learning paradigm. BEiT (Bao et al., 2021) pioneered masked prediction of discrete visual tokens, while MAE (He et al.,

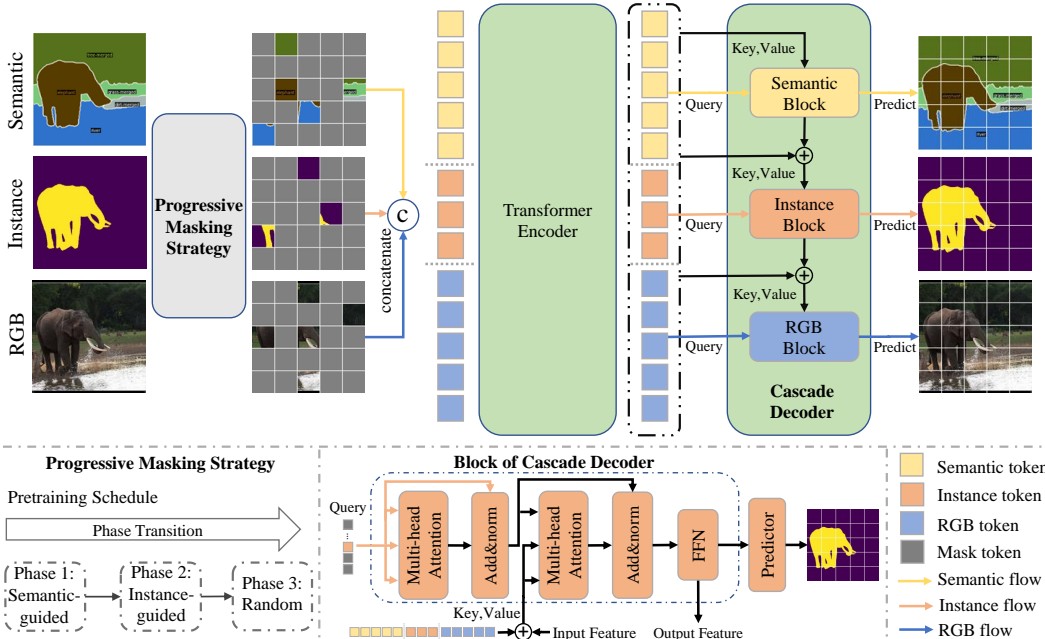

Figure 2: CFMAE pre-training framework. Multi-granular data (RGB, Instance, Semantic masks) is first masked by Progressive Masking and then concatenated, and fed to a transformer encoder. Encoded tokens subsequently flow into a cascaded decoder with three task-specific blocks. Each block is a standard Transformer decoder block, composed of self-attention, cross-attention, and feed-forward network layers. We use linear layers as the final predictor. As training progresses, the masking strategy transitions from semantic-guided masking to instance-guided masking, and finally to random masking to build hierarchical visual representations.

2022) introduced an influential asymmetric encoder-decoder architecture with a high masking ratio for predicting raw pixels. Subsequent works like SimMIM (Xie et al., 2022b) (lighter decoder), iBOT (Zhou et al., 2022) (online tokenizer), and MaskFeat (Wei et al., 2022) (HOG features) further refined MIM. Other approaches explored reconstructing deep features (Wang et al., 2023; Ren et al., 2023), frequencies (Xie et al., 2022a; Xiang et al., 2025), or learning from corrupted (Fang et al., 2022) and noisy images (You et al., 2024; Xiang et al., 2024). MultiMAE (Bachmann et al., 2022) extended this paradigm to multi-modal inputs, using a parallel decoder to reconstruct different modalities simultaneously. Beyond random masking, structured strategies have been investigated: MST (Li et al., 2021) and AttMask (Kakogeorgiou et al., 2022) used attention maps; ADIOS (Shi et al., 2022) and AutoMAE (Chen et al., 2023a) employed adversarial mask generation; SemMAE (Li et al., 2022a) leveraged semantic parts; and UnMAE (Li et al., 2022b) introduced Uniform Masking. These methods use fixed strategies with specific inductive biases, unlike our progressive masking, which learns hierarchical representations by evolving the masking focus from high-level semantics to low-level details.

## 2.2 HIERARCHICAL REPRESENTATION LEARNING

Learning to capture multi-scale and multi-level features, from low-level textures to high-level semantics, is a core objective in computer vision. Swin Transformer (Liu et al., 2021)introduces structural biases into ViT through shifted windows and a hierarchical backbone. For specific tasks, H-ViT (Ghahremani et al., 2024) also constructs a hierarchical Transformer architecture. In terms of architecture-agnostic approaches, Hiera (Ryali et al., 2023) demonstrates that effective hierarchical representations can be obtained simply by learning masked autoencoding on multi-scale features. HGCLIP (Xia et al., 2023) aligns visual features with a predefined class hierarchy. Furthermore, Zhang & Maire (2020) drives hierarchical learning through self-supervised region grouping. Examples of applying the coarse-to-fine strategy for specific downstream tasks include MIMO-UNet (Cho et al., 2021) for image deblurring and CF-ViT (Chen et al., 2023b) for accelerating inference. Unlike

these methods, our model, through a unified progressive masking and cascaded decoder, is the first to establish the coarse-to-fine philosophy as a general pre-training paradigm for hierarchical representation learning.

# 3 METHOD

This section details the proposed CFMAE framework. A prerequisite for our coarse-to-fine pre-training is a dataset with aligned annotations across different levels of granularity. We construct this large-scale dataset by generating high-quality pseudo-labels for instance and semantic segmentation on ImageNet-1K. The detailed methodology for this dataset construction is provided in Appendix B.

As illustrated in Figure 2, our CFMAE architecture is then built upon four core components that leverage this data: multi-granular inputs, a shared encoder, a cascaded decoder, and a top-down progressive masking strategy. Each of these components is detailed in the following subsections.

## 3.1 MULTI-GRANULAR INPUTS AND SHARED ENCODER

Our framework utilizes data at three distinct granularities: RGB images ($\mathbf{I}_{rgb} \in \mathbb{R}^{H \times W \times 3}$), instance segmentation masks ($\mathbf{I}_{ins} \in \mathbb{Z}^{H \times W}$), and semantic segmentation masks ($\mathbf{I}_{sem} \in \mathbb{Z}^{H \times W}$), where $H, W$ are the image dimensions. Each granularity of input $\mathbf{I}_m$ for $m \in \{rgb, ins, sem\}$ is first divided into a sequence of $N$ non-overlapping patches. These patches are then flattened and mapped to $D$-dimensional embeddings through granularity-specific linear projection layers, resulting in a sequence of tokens $\mathbf{Z}_m = \{\mathbf{z}_m^1, \ldots, \mathbf{z}_m^N\} \in \mathbb{R}^{N \times D}$. Following the application of our progressive masking strategy (detailed in Section 3.3), which yields a binary mask, only the visible tokens of each granularity are selected. The visible tokens across all granularities are then concatenated into a single sequence $\mathbf{Z}^{vis}$ and fed into a shared Vision Transformer (Dosovitskiy et al., 2021) encoder:

$$\mathbf{H}_{enc} = \text{Encoder}(\mathbf{Z}^{vis}), \tag{1}$$

where $\mathbf{H}_{enc} \in \mathbb{R}^{N_{vis} \times D}$ is the sequence of encoded visible tokens, with $N_{vis}$ being the number of visible tokens. This unified representation serves as a comprehensive foundation for the subsequent hierarchical feature refinement in the cascaded decoder.

## 3.2 CASCADED DECODER ARCHITECTURE

Our framework features a cascaded decoder that progressively refines features across three sequential task-specific blocks, indexed by $k \in \{1, 2, 3\}$. These blocks correspond to the reconstruction tasks $t \in \{S, I, R\}$ (Semantic, Instance, RGB), respectively, aligning with our top-down masking strategy. As depicted in Figure 2, the input to the decoder is the sequence of encoded visible tokens $\mathbf{H}_{enc} \in \mathbb{R}^{N_{vis} \times D}$. Each decoder block $k$ is a standard Transformer decoder block, composed of self-attention, self-attention, feed-forward network layers. We use simple linear layers for the final prediction. For a given task $t$, the inputs to the decoder block $k$ are defined as follows:

**Query ($\mathbf{Q}^k$):** The encoded visible tokens $\mathbf{H}_{enc}$ are first combined with the mask tokens to get full tokens $\mathbf{H} \in \mathbb{R}^{3N \times D}$. Since the tokens for different tasks maintain fixed positions within this sequence, the query specific to task $t$ is obtained by slicing $\mathbf{H}$ along the sequence dimension based on pre-defined start and end indices $(i_t, j_t)$ as $\mathbf{Q}^k = \mathbf{H}[i_t : j_t] \in \mathbb{R}^{N \times D}$.

**Key ($\mathbf{K}^k$) and Value ($\mathbf{V}^k$):** The key and value are formed by fusing the full token sequence $\mathbf{H}$ with the output features from the preceding block, $\mathbf{F}^{k-1} \in \mathbb{R}^{N \times D}$ (initialized as $\mathbf{F}^0 = \mathbf{0}$). This is achieved via element-wise addition of tokens at corresponding positions: $\mathbf{K}^k = \mathbf{V}^k = \mathbf{H} \oplus \mathbf{F}^{k-1}$.

The output feature of each task block, $\mathbf{F}^k \in \mathbb{R}^{N \times D}$, is obtained by applying the decoder block:

$$\mathbf{F}^k = \text{DecoderBlock}^k(\mathbf{Q}^k, \mathbf{K}^k, \mathbf{V}^k). \tag{2}$$

Finally, a task-specific linear predictor generates the reconstruction for task $t$:

$$\hat{\mathbf{I}}^t = \text{Predictor}^t(\mathbf{F}^k), \tag{3}$$

where $\hat{\mathbf{I}}^t$ is the reconstructed output for the corresponding task. In contrast to parallel architectures, this progressive refinement allows each stage to build upon the last, enforcing a coarse-to-fine information flow that is crucial for learning hierarchical representations.

### 3.3 TOP-DOWN PROGRESSIVE MASKING STRATEGY

Our top-down progressive masking strategy facilitates hierarchical representation learning through three pre-training phases using the following generated masks:

- Semantic-guided mask ($\mathbf{M}_S$): Applies random masking within each semantic region, with the number of masked patches allocated to each region being proportional to its relative area.

- Instance-guided mask ($\mathbf{M}_I$): Guides masking based on instance information, prioritizing the occlusion of object regions over the background.

- Random mask ($\mathbf{M}_R$): Applies standard uniform random masking across the image with no structural guidance.

The final mask $\mathbf{M} \in \{0, 1\}^{3N}$ is generated from an intermediate score map $\mathbf{M}_{score}$. This score map is a weighted combination of the binary masks $\mathbf{M}_S, \mathbf{M}_I, \mathbf{M}_R \in \{0, 1\}^N$, controlled by coefficients $\alpha_I$ and $\alpha_S$:

$$\mathbf{M}_{score} = (1 - \alpha_I - \alpha_S)\mathbf{M}_R + \alpha_I\mathbf{M}_I + \alpha_S\mathbf{M}_S. \tag{4}$$

The final binary mask $\mathbf{M}$ is then obtained by masking $r \cdot 3N$ patches corresponding to higher scores in $\mathbf{M}_{score}$, where $r$ is the the overall masking ratio. The coefficients $\alpha_I, \alpha_S$ (where $0 \leq \alpha_I, \alpha_S$ and $\alpha_I + \alpha_S \leq 1$) are dynamically adjusted during pre-training to smoothly transition the masking focus from semantic-guided to instance-guided and finally to random masking. This progression creates a curriculum that guides the model to construct increasingly sophisticated visual representations. Detailed formulations for generating $\mathbf{M}_S$, $\mathbf{M}_I$, and $\mathbf{M}_R$, and the schedule for $\alpha_I, \alpha_S$ are provided in the Appendix C.

### 3.4 TRAINING OBJECTIVES AND LOSS FUNCTIONS

Our framework is optimized with a multi-task objective to learn a comprehensive representation across distinct granularity levels. The overall loss combines three reconstruction losses, each aligned with a stage in our cascaded decoder:

$$\mathcal{L}_{\text{total}} = \lambda_S\mathcal{L}_{\text{S}} + \lambda_I\mathcal{L}_{\text{I}} + \lambda_R\mathcal{L}_{\text{R}}, \tag{5}$$

where $\mathcal{L}_{\text{S}}$ and $\mathcal{L}_{\text{I}}$ are the cross-entropy losses for semantic and instance mask prediction, respectively, while $\mathcal{L}_{\text{R}}$ is the mean squared error for RGB image reconstruction. The coefficients $\lambda_{\text{S}}$, $\lambda_{\text{I}}$, and $\lambda_{\text{R}}$ are weighting factors that balance the contribution of each task, guiding the model to learn a coarse-to-fine hierarchical visual representation.

## 4 EXPERIMENTS

This section outlines our experimental validation. We first present the main experimental results, benchmarking CFMAE against state-of-the-art methods and evaluating its robustness. Following this, we conduct comprehensive ablation studies to dissect the contributions of key components within our framework. Finally, qualitative visualizations are provided to intuitively demonstrate the efficacy of our proposed approach. Detailed experimental settings for the pre-training and downstream task evaluations can be found in the Appendix D. All experiments were conducted on a server with $8\times$ NVIDIA Tesla-A100 GPUs.

### 4.1 MAIN RESULTS

**Image Classification.** We first evaluate our method on ImageNet-1K, comparing its fine-tuning top-1 accuracy against key baselines and state-of-the-art methods. Alongside the foundational MAE baseline, MultiMAE serves as a particularly strong counterpart, as it also leverages multi-modal data but employs a parallel decoder and a simple random masking strategy.

As shown in Table 1, our CFMAE achieves fine-tuning accuracies of 83.7% and 84.2% after 400 and 1600 pre-training epochs, respectively. This represents a significant improvement over both the baseline MAE (+0.8%/+0.6%) and MultiMAE (+1.0%/+0.9%). Notably, MultiMAE underperforms the simpler MAE, suggesting that its parallel processing of modalities fails to effectively integrate

Table 1: Performance comparison of self-supervised methods on ImageNet-1K. We report fine-tuning Top-1 accuracy (%) of ViT-B. The input size is $224 \times 224$. PT Cost is the relative time to MAE (400 epochs), which is taken as 1.0. * indicates results reproduced using the official code. † means using multi-granular data as input for fine-tuning.

| Method | Model | Modality | Masking | PT Epoch | PT Cost | Acc. |
|---|---|---|---|---|---|---|
| DINO (Caron et al., 2021) | ViT-B | RGB | – | 300 | – | 82.8 |
| BEiT (Bao et al., 2021) | ViT-B | RGB | Random | 800 | ~7.0× | 83.2 |
| MAE (He et al., 2022)* | ViT-B | RGB | Random | 400 | ~1.0× | 82.9 |
| MAE (He et al., 2022) | ViT-B | RGB | Random | 1600 | ~4.0× | 83.6 |
| iBOT (Zhou et al., 2022) | ViT-B | RGB | Random | 1600 | ~5.7× | 84.0 |
| CAE (Chen et al., 2024) | ViT-B | RGB | Random | 800 | ~4.6× | 83.6 |
| MaskFeat (Cheng et al., 2022) | ViT-B | RGB | Random | 1600 | ~20.1× | 84.0 |
| SemMAE (Li et al., 2022a) | ViT-B | RGB | Semantic | 800 | – | 83.3 |
| ConMIM (Yi et al., 2023) | ViT-B | RGB | Random | 800 | ~4.4× | 83.7 |
| MIRL (Huang et al., 2023) | ViT-B | RGB | Random | 800 | – | 84.1 |
| ROPIM (Haghighat et al., 2024) | ViT-B | RGB | Random | 800 | ~10.4× | 84.0 |
| MFM (Xie et al., 2022a) | ViT-B | RGB/Frequency | Random | 300 | ~1.1× | 83.1 |
| MultiMAE* (Bachmann et al., 2022) | ViT-B | RGB/Dep./Sem. | Random | 400 | ~1.3× | 82.7 |
| MultiMAE (Bachmann et al., 2022) | ViT-B | RGB/Dep./Sem. | Random | 1600 | ~5.2× | 83.3 |
| CFMAE | ViT-B | RGB/Inst./Sem. | Progressive | 400 | ~1.3× | 83.7 |
| CFMAE | ViT-B | RGB/Inst./Sem. | Progressive | 1600 | ~5.2× | **84.2** |
| CFMAE† | ViT-B | RGB/Inst./Sem. | Progressive | 1600 | ~5.2× | **84.4** |

Table 2: Object detection and instance segmentation results on the COCO dataset, with evaluation metrics of $AP^b$ (%) and $AP^m$ (%). The pre-trained ViT-B backbone is integrated into the Mask R-CNN framework for end-to-end fine-tuning.

| Method | Model | PT Epoch | $AP^b$ | $AP^m$ |
|---|---|---|---|---|
| BEiT | ViT-B | 800 | 35.6 | 32.6 |
| MAE | ViT-B | 1600 | 48.3 | 42.5 |
| CAE | ViT-B | 800 | 49.8 | 43.9 |
| iBOT | ViT-B | 1600 | 48.3 | 42.7 |
| ConMIM | ViT-B | 800 | 47.8 | 42.5 |
| MIRL | ViT-B | 800 | 49.3 | 43.7 |
| MultiMAE | ViT-B | 1600 | 48.1 | 42.2 |
| CFMAE | ViT-B | 1600 | **50.1** | **44.1** |

Table 3: Semantic segmentation results on the ADE20K dataset, with evaluation metrics of mIoU. The pre-trained ViT-B backbone is integrated with UperNet for end-to-end fine-tuning.

| Method | Model | PT Epoch | mIoU |
|---|---|---|---|
| BEiT | ViT-B | 800 | 47.1 |
| MAE | ViT-B | 1600 | 48.1 |
| MaskFeat | ViT-B | 1600 | 48.8 |
| SemMAE | ViT-B | 800 | 46.3 |
| ConMIM | ViT-B | 1600 | 46.0 |
| ROPIM | ViT-B | 300 | 48.5 |
| MultiMAE | ViT-B | 1600 | 47.8 |
| CFMAE | ViT-B | 1600 | **49.1** |

hierarchical information. In contrast, CFMAE's superior performance validates that our coarse-to-fine framework, with its cascaded decoder and progressive masking, successfully learns more powerful hierarchical representations. Furthermore, we explored fine-tuning on ImageNet-1K using multi-granular data as input, achieving a final accuracy of 84.4%, which is competitive with current state-of-the-art methods. This indicates that within our framework, the different data granularities can also synergistically boost downstream task performance.

In terms of computational cost, CFMAE's training time is nearly identical to MultiMAE's and only about 1.3 times that of MAE, primarily due to the increased number of input tokens. The overhead from the cascaded decoder and progressive mask generation is negligible. Crucially, our 400-epoch model already surpasses the performance of MAE's 1600-epoch model (83.7% vs. 83.6%). This demonstrates that CFMAE not only achieves higher accuracy but does so more efficiently, constructing rich hierarchical representations in a fraction of the training time.

**Object Detection and Instance Segmentation.** To evaluate the transferability of our approach to dense downstream tasks, we conduct experiments on object detection and instance segmentation

using the COCO dataset. As shown in Table 2, our pre-trained model achieves significant gains over key baselines. Specifically, CFMAE outperforms MAE by +1.8 AP$^b$ and +1.6 AP$^m$, and surpasses MultiMAE by +2.0 AP$^b$ and +1.9 AP$^m$ in object detection and instance segmentation, respectively. These substantial improvements underscore that our coarse-to-fine pre-training framework effectively enhances the model's ability to capture the hierarchical features crucial for complex dense prediction tasks.

**Semantic Segmentation.** We also evaluate our method on semantic segmentation using the ADE20K dataset, with mIoU as the evaluation metric. The results, summarized in Table 3, show that our approach consistently surpasses the performance of competing methods. These findings further highlight the effectiveness of our framework in capturing multi-level semantic information, enabling superior generalization across diverse segmentation tasks.

**Robustness Evaluation.** We assess the robustness of our methods across various out-of-distribution (OOD) ImageNet benchmarks, including ImageNet-A (Hendrycks et al., 2021b), ImageNet-R (Hendrycks et al., 2021a), ImageNet-Sketch (Wang et al., 2019), and ImageNet-C (Hendrycks & Dietterich, 2019). Top-1 accuracy is the primary evaluation metric for all datasets, except for ImageNet-C, where we report mean corruption error (mCE). To derive the final robustness score, we calculate $1 - \text{mCE}$ and take the average across all tested datasets. As shown in Table 4, our method demonstrates superior robustness compared to other approaches. We achieve improvements across all four OOD datasets, with the most significant gains observed on ImageNet-R and ImageNet-Sketch. Specifically, our model shows an average score increase of 1.8% and 0.9% over MAE and MultiMAE, highlighting that our framework helps the model learn more robust visual representations, significantly enhancing its robustness against OOD data.

Table 4: Robustness results of ViT-B on various ImageNet OOD variants. Top-1 accuracy is reported for all datasets, except for ImageNet-C, where the mean corruption error (mCE) is the evaluation metric. And (1-mCE) is used to calculate the average score.

| Method | IN-A | IN-R | IN-S | IN-C ↓ | Score |
|---|---|---|---|---|---|
| MAE | **35.9** | 48.3 | 34.5 | 51.7 | 41.8 |
| MFM | 32.7 | 48.6 | 34.8 | **47.5** | 42.2 |
| MultiMAE | 33.2 | 49.9 | 37.2 | 49.6 | 42.7 |
| CFMAE | 35.2 | **50.6** | **37.4** | 48.8 | **43.6** |

## 4.2 ABLATION STUDIES

In this section, we perform ablation studies to evaluate the key components of our framework. All models are pre-trained for 400 epochs using ViT-B. We first conduct a step-by-step component analysis. Starting from the MultiMAE baseline, we incrementally add our proposed components. The results are presented in Table 5. This incremental performance gain at each step strongly verifies the effectiveness and synergistic nature of our proposed components. Then, we further investigate the hyper-parameters and design choices for different components, including input tokens, masking strategy, decoder design, input modality, and loss weight. The results are summarized in Table 6.

Table 5: Component analysis of CFMAE on ImageNet-1K. Starting from a MultiMAE baseline, we incrementally add our proposed components.

| Configuration | Top-1 |
|---|---|
| Baseline (MultiMAE) | 82.7 |
| + Our Dataset (R+I+S) | 83.0 (+0.3) |
| + Cascaded Decoder | 83.3(+0.3) |
| + Progressive Masking (Ours) | **83.7**(+0.4) |

**Input Tokens.** We use the number of input tokens as a proxy for the mask ratio. Given that the total number of tokens across the three modalities is set to $196 \times 3$ by default, an input token count of 98/147/196 corresponds to a mask ratio of 0.833/0.75/0.667. Table 6a demonstrates that in our framework, a higher mask ratio (fewer input tokens) facilitates the model in capturing both modality-specific features and cross-modal interactions. This leads to improved representation learning while also enhancing training efficiency.

Table 6: CFMAE ablation experiments on ImageNet-1K. The fine-tuning Top-1 accuracy (%) is reported. The default settings include 98 for the number of input tokens, progressive masking with the order of SG→IG→RD, a cascaded decoder with cross-attention and task sequence of S→I→R, R+I+S for the input modality, and $\lambda_S = \lambda_I = \lambda_R = 1$ for loss weight. The selected settings are underlined.

(a) Input Tokens.

| Number | Top-1 |
|--------|-------|
| 98 | **83.7** |
| 147 | 83.7 |
| 196 | 83.5 |

(b) Single Masking strategy.

| Type | Top-1 |
|------|-------|
| RD | 83.3 |
| IG | **83.5** |
| SG | 83.4 |

(c) Masking order.

| Type | Top-1 |
|------|-------|
| IG→SG→RD | 83.5 |
| RD→IG→SG | 83.5 |
| SG→IG→RD | **83.7** |

(d) Decoder Design.

| Type | Top-1 |
|------|-------|
| Parallel | 83.3 |
| w/o CA | 83.2 |
| R→I→S | 83.5 |
| S→I→R | **83.7** |

(e) Input modality.

| Modality | Top-1 |
|----------|-------|
| RGB | 82.9 |
| R+S | 83.2 |
| R+I | 83.1 |
| R+I+S | **83.3** |

(f) Loss weight.

| $\lambda_S, \lambda_I, \lambda_R$ | Top-1 |
|-----------------------------------|-------|
| 1, 1, 1 | **83.7** |
| 1, 1, 2 | 83.7 |
| 1, 2, 1 | 83.4 |
| 2, 1, 1 | 83.5 |

Figure 3: Predictions of CFMAE on masked multi-granular data. All the tested images are from the ImageNet-1K validation set and masked with the random masking strategy.

**Mask strategy.** To verify the effectiveness of our proposed masking strategy, we first evaluate the three core masking approaches, Random Masking (RD), Instance-Guided Masking (IG), and Semantic-Guided Masking (SG) individually. The results in Table 6b demonstrate that both IG and SG strategies contribute positively to representation learning compared to RD. Building on this, to assess the impact of dynamically sequencing these strategies, we test our progressive masking approach with different orders. As evidenced in Table 6c, our progressive masking strategy demonstrates superior efficacy over static random masking in facilitating hierarchical visual representations. The top-down masking order(SG→IG→RG), well-aligned with our cascaded decoder architecture, yields further performance gains. Our progressive masking scheme enables the model to progressively construct visual features through semantic abstraction hierarchies, beginning with high-level semantic concepts and gradually refining localized detail representations.

**Decoder Desgin.** We evaluate the effectiveness of cascaded decoders within our framework, and further investigate the impacts of cross-attention mechanisms and task sequencing in reconstruction objectives. As demonstrated in Table 6d, the cascaded decoder architecture significantly enhances the hierarchical construction of visual representations compared to the parallel decoder. The task sequence, starting with semantic mask prediction, followed by instance mask prediction, and ending with RGB image reconstruction, effectively supports hierarchical feature development. Notably, cross-attention plays a critical role in the decoder design by facilitating effective cross-modal interaction.

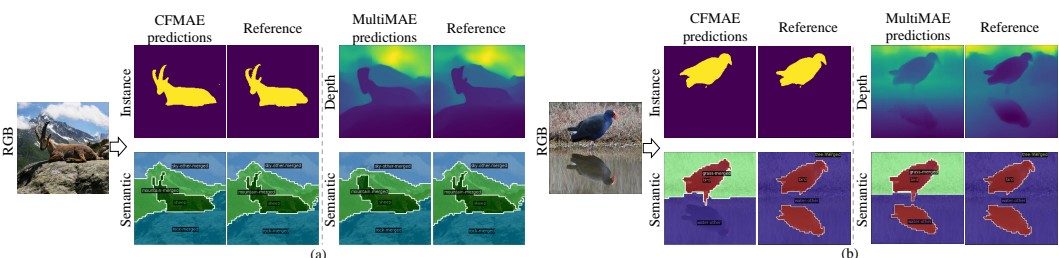

Figure 4: Single-modal prediction of CFMAE and MultiMAE on the ImageNet-1K validation set.

**Input modality.** We conduct an ablation study to isolate the contribution of each data granularity. Here, we use random masking because progressive masking requires the presence of all granularities. The results presented in Table 6e clearly illustrate that data from each distinct granularity positively contributes to the visual model's representation learning. Moreover, these granularities exhibit a synergistic effect, collectively enhancing the model's capacity to learn more robust and superior visual representations.

**Loss weight.** We investigate the impact of different weighting schemes for the semantic ($\lambda_S$), instance ($\lambda_I$), and RGB ($\lambda_R$) reconstruction losses. As shown in Table 6f, we observe that when RGB reconstruction is the dominant task (e.g., $\lambda_R \geq \lambda_S, \lambda_I$), downstream task performance is generally strong, likely because RGB images implicitly contain the richest information. For simplicity, we adopt equal weights ($\lambda_S = \lambda_I = \lambda_R = 1$) as our default setting.

### 4.3 VISUALIZATIONS

To highlight the improvements in representation learning achieved by our dataset and framework, we visualize the reconstruction results of our pre-trained auto-encoder across different data modalities. All images are from the ImageNet-1K validation set. Figure 3 illustrates the reconstruction capabilities of CFMAE on masked multi-granular data. As shown, our model effectively reconstructs each granular data under different input conditions. Thanks to the interactivity and complementarity between the multi-granular dataset we have constructed, our model performs well in predicting fine details. Moreover, in some cases, it can even correct errors in the pseudo-labels, as seen in the instance prediction in Figure 3b. We further explore using single-modal data to predict data from other modalities, *e.g.* leveraging fine-grained RGB images to predict coarse-grained instance and semantic masks. The results shown in Figure 4 highlight the transferability of our method across different modalities. Compared to MultiMAE, our model not only delivers superior performance in predicting finer details but also demonstrates greater robustness, *e.g.* being unaffected by reflections of birds in water in Figure 4b. More visualization results can be found in Appendix E.

### 5 CONCLUSION

In this paper, we propose CFMAE, a novel coarse-to-fine vision pre-training framework that hierarchically integrates RGB images, instance masks, and semantic masks through a cascaded decoder and progressive masking strategy. By adhering to a strict coarse-to-fine principle, our method builds hierarchical representations from global semantics to fine-grained details, achieving state-of-the-art performance across classification, detection, and segmentation tasks. Furthermore, the large-scale, multi-granular pseudo-labels dataset we constructed provides a valuable resource for future research in hierarchical representation learning.

**Limitations.** While CFMAE shows promising results, future work could explore several directions. First, despite its multi-granular scalability, the current CFMAE integrates only RGB, instance, and semantic masks. Incorporating finer-grained segmentation or broader visual modalities could further enhance hierarchical representation learning. Second, to inject richer semantic knowledge into the hierarchy, a feature alignment module could be integrated at the initial decoder stages. By aligning the model's internal features with powerful external embeddings (e.g., from CLIP (Radford et al., 2021)), this approach could significantly boost the final hierarchical representations. These aspects are left for future exploration.

**Reproducibility Statement.** The Key source code of MsTok has been included in the supplementary materials. All execution configurations, relevant parameters, and tratectory sampling implementations are provided within the associated scripts and data projects, facilitating reproducibility of the results. The core CFMAE source code (cascaded decoder, progressive masking) is included in the supplementary materials. All hyper-parameters for pre-training and downstream fine-tuning are specified in the main paper and Appendix (Tables 7, 8, 9, 10). The dataset construction (instance/semantic pseudo-label generation) is fully described in Appendix B. Together these resources enable faithful reproduction of our results.

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

## APPENDIX

In this appendix, we first provide the statement about the use of large Language Models(LLMS) in Section A. We then show the details of how to construct the multi-granular pseudo-label dataset based on ImageNet-1K in Section B. Then, we present the detailed formulations of Progressive Masking Strategy in Section C. Subsequently, we show more insightful visualization results on the proposed dataset and single-modal/multi-modal predictions in Section D. Finally, we provide the detailed experimental settings for the pre-training and fine-tuning stage in Section E.

## A  THE USE OF LARGE LANGUAGE MODELS(LLMS)

We acknowledge the use of large language models (LLMs) as writing assistants only for grammaticaand style. LLMs are not employed in the core research methodology, experimental design, dataanalysis, or generation of research findings presented in this paper. All textual content has beerrigorously reviewed and verified by the authors to ensure accuracy and authenticity of the researchcontributions.

## B  DATASET CONSTRUCTION

To facilitate hierarchical visual representation learning, we construct a large-scale multi-granular visual dataset that augments ImageNet with multi-level segmentation annotations. Building upon the complete ImageNet-1K dataset (1.28M training images), we leverage SAM-based methods (Kirillov et al., 2023; Ke et al., 2024; Zou et al., 2024; Ren et al., 2024) to generate high-quality instance segmentation masks for individual objects, as well as corresponding semantic segmentation masks for entire image regions. Each image in our dataset is accompanied by precisely aligned instance-level and semantic-level annotations, enabling models to progressively learn scene-level semantics, object-level structures, and pixel-level details.

To generate high-quality instance-level segmentation annotations, we develop a two-stage pipeline referring to Grounded SAM (Ren et al., 2024). For each image in ImageNet, we first utilize Grounded DINO (Liu et al., 2023) to detect all relevant objects based on the provided text prompts, where we simply use category names as the prompts for efficient processing. The detected objects are then cropped and processed as input to the SAM model to obtain instance masks. To further enhance the mask quality, we employ HQ-SAM (Ke et al., 2024) as the second-stage segmentation model, which significantly improves the mask details, particularly around object boundaries. During the dataset construction process, we adopt an adaptive confidence thresholding strategy. Starting with a high confidence threshold, we generate initial masks for images where target objects are clearly visible. For images with few or no masks, we gradually lower the confidence threshold to ensure comprehensive coverage while maintaining annotation quality. This approach effectively prevents the

generation of excessive irrelevant masks within a single image, thereby ensuring the overall quality of our instance-level annotations.

To generate high-quality semantic segmentation annotations for our ImageNet-based dataset, we leverage another SAM-based framework SEEM (Zou et al., 2024). Previous work, MultiMAE (Bachmann et al., 2022), utilized the Mask2Former (Cheng et al., 2022) model with a Swin-S (Liu et al., 2021) backbone trained on COCO (Lin et al., 2014) dataset. In contrast, our approach employs a more advanced SAM-based methodology, which demonstrates superior boundary accuracy, better handling of complex scenes, and enhanced ability to segment fine-grained details, compared to prior techniques. Moreover, it exhibits stronger generalization capability across diverse object appearances and scenarios, which is particularly crucial for the wide variety of images in ImageNet. To ensure optimal segmentation quality, we employ the large-size model of the SEEM, which provides a more comprehensive semantic understanding and more precise mask generation compared to smaller architectures. Following the COCO categorical structure, we define a total of 133 classes, comprising 80 thing classes for countable objects and 53 stuff classes for uncountable background elements. This rich categorical system enables our model to capture both foreground objects and contextual information effectively.

We provide visualizations of our dataset in the Appendix D.1.

## C    Detailed formulations of Progressive Masking

The progressive masking strategy guides the model through three distinct learning phases. Below we detail each phase of our masking strategy and explain how they collectively contribute to hierarchical representation learning.

**Random Masking for Local Feature Understanding.** In this phase, we employ random masking to establish foundational visual understanding. Given input data from different modalities divided into N patches, we apply independent random masking for each modality. For each modality $m \in \{rgb, ins, sem\}$ where $rgb, ins, sem$ means RGB/Instance/Semantic, its masking ratio $r_m$ is sampled from a Dirichlet distribution following MultiMAE (Bachmann et al., 2022), with a total masking ratio of $r$. Therefore, the masking process can be described as:

$$\mathbf{M}_R^m = f_{rand}(r_m) \in \{0, 1\}^N, \tag{6}$$

where $f_{rand}$ is a random select function, $\mathbf{M}_R^m$ denotes the binary masking matrix for modality $m$, with 1 indicating masked positions. Note that $|\mathbf{M}_R^m| = \lfloor r_m N \rfloor$, where $|\cdot|$ represents the matrix norm of the mask, and $\lfloor \cdot \rfloor$ represents the floor function.

**Instance-guided Masking for Object-level Understanding.** In this phase, we transition to instance-guided masking to promote object-centric learning. Given the instance masks $\mathbf{I}_{ins}$, we distribute the masked patches with emphasis on object regions. Let $\Omega_{obj}$ and $\Omega_{bg}$ denote the sets of patches belonging to object regions and background regions, respectively. The masking process can be formulated as:

$$\mathbf{M}_I^m = f_{ins}(\mathbf{I}_{ins}, r_m, \alpha) \in \{0, 1\}^N, \tag{7}$$

where $f_{ins}$ is our instance-guided masking function, and $\alpha$ controls the distribution of masks between object and background regions. Specifically, $f_{ins}$ includes: 1) identifying complete object instances from $\mathbf{I}_{ins}$; 2) selecting all or a subset of instances based on their size and spatial significance; 3) generating masks by randomly and separately selecting masked patches in object and background regions according to the ratio $\alpha$, which means

$$|\mathbf{M}_I^m \cap \Omega_{obj}| = \lfloor \alpha r_m N \rfloor, \tag{8}$$
$$|\mathbf{M}_I^m \cap \Omega_{bg}| = r_m N - \lfloor \alpha r_m N \rfloor.$$

We assign $\alpha > 0.5$ to prioritize masking object regions, ensuring that a larger portion of masked patches is allocated to object regions while maintaining some masking in the background for contextual learning.

**Semantic-guided Masking for Scene-level Perception.** In this phase, we introduce semantic-guided masking based on the semantic regions. Given semantic masks $\mathbf{I}_{sem}$ with $C$ classes, we assign different masking weights to different semantic regions. Let $\Omega_c$ denote the set of patches belonging

to semantic class $c$, and $w_c$ represent the corresponding class weight. The masking process can be formulated as:

$$\mathbf{M}_S^m = f_{sem}(\mathbf{I}_{sem}, r_m, \boldsymbol{w}) \in \{0,1\}^N \tag{9}$$

where $f_{sem}$ is the semantic-guided masking function, and $\boldsymbol{w}$ are the class-specific weights. Specifically, $f_{sem}$ includes: 1) grouping patches according to their semantic labels to regions; 2) calculating the number of masked patches for each semantic region based on both region importance weights and region sizes; 3) randomly selecting patches within each region. For semantic class $c$, the number of masked patches is determined by:

$$|\mathbf{M}_S^m \cap \Omega_c| = \left\lfloor r_m N \cdot \frac{w_c |\Omega_c|}{\sum_{k=1}^C w_k |\Omega_k|} \right\rfloor . \tag{10}$$

When all weights are equal, the masking distribution is solely determined by the region sizes, effectively becoming region-wise random masking. We adopt this configuration in our framework for simplicity.

**Progressive Training Schedule.** To facilitate seamless transitions across three pre-training phases, we propose a progressive training schedule, that enables hierarchical knowledge accumulation while maintaining training stability. The transition between phases is controlled by the mixing coefficients $\alpha, \alpha_S$, formulated as:

$$\mathbf{M}^m = f_t((1 - \alpha_I - \alpha_S)\mathbf{M}_R^m + \alpha_I \mathbf{M}_I^m + \alpha_S \mathbf{M}_S^m, r_m), \tag{11}$$

where $\mathbf{M}^m \in \{0,1\}^N$ is the final mask for modality $m$, $0 \leq \alpha_I, \alpha_S, \alpha_I + \alpha_S \leq 1$, and $f_t$ prioritizes positions with higher value when selecting the final masked patches. During pre-training, we dynamically adjust the values of $\alpha_I$ and $\alpha_S$ to ensure smooth transitions of masking strategies from the first phase to the third phase. We show the setting of $\alpha_I$ and $\alpha_S$ in Figure 5. This progressive

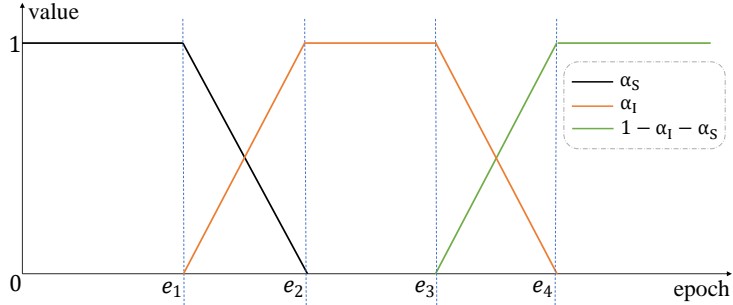

Figure 5: The variation curves of $\alpha_I$ and $\alpha_S$ during the training process.

approach enables the model to build increasingly sophisticated visual representations, from local patterns to object structures and semantic relationships, leading to more robust and hierarchical visual representations.

# D VISUALIZATION RESULTS

## D.1 ILLUSTRATION OF OUR DATASET

In Figure 6, we randomly display images representing various categories from our dataset. Each cell is structured in three columns: the first column presents the original RGB image, followed by its corresponding instance masks and semantic masks. The visualization effectively illustrates our dataset's exceptional segmentation precision across diverse classes of images.

## D.2 RECONSTRUCTION RESULTS ON RANDOMLY MASKED MULTI-MODAL DATA.

Figure 7 further illuminates our framework's generative capabilities when addressing randomly masked multi-modal data from the unseen ImageNet validation set. Despite the diverse object categories, our method adeptly reconstructs RGB images, instance masks, and semantic masks, demonstrating remarkable precision in object boundary delineation and semantic comprehension.

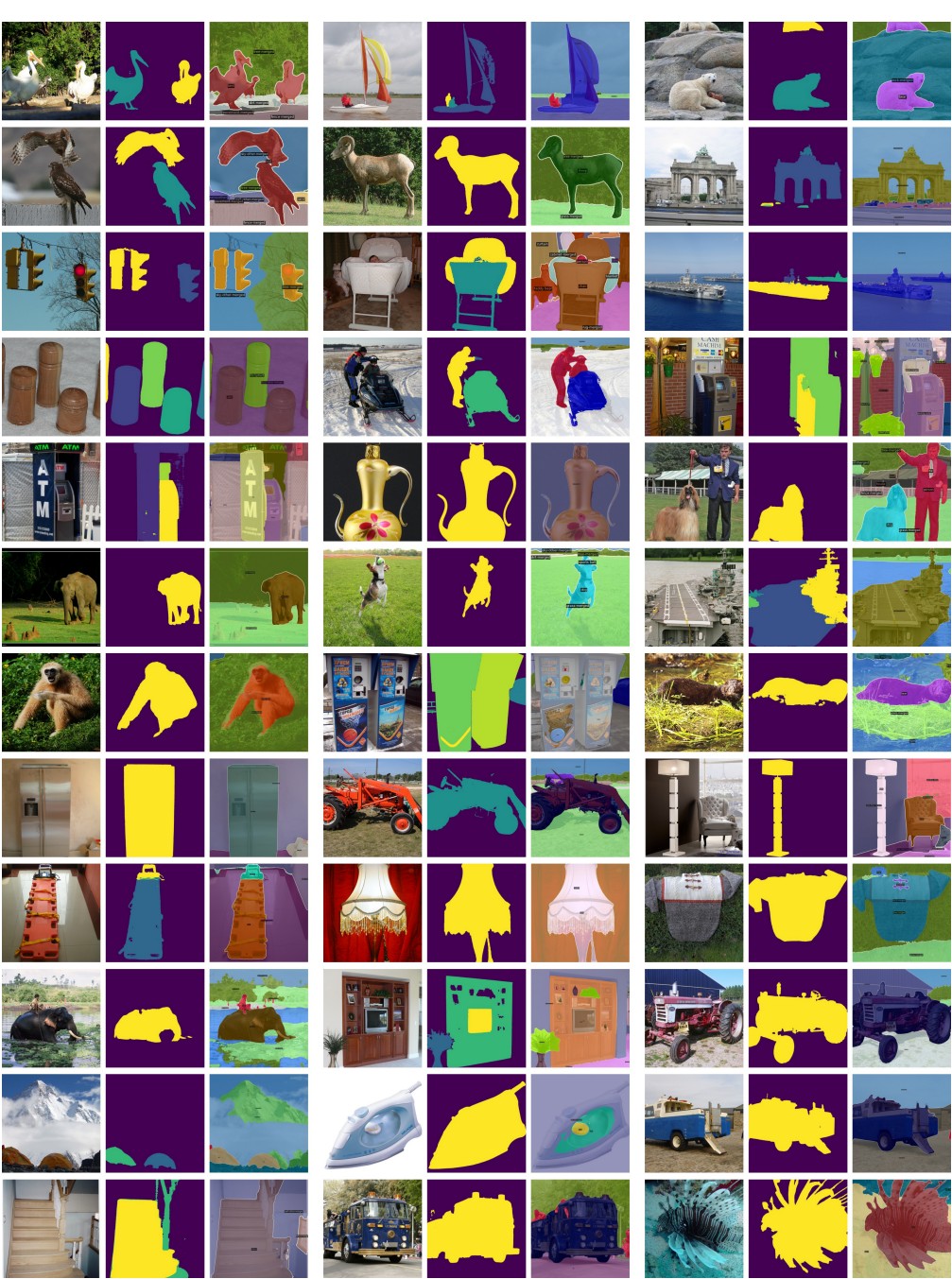

Figure 6: Illustration of the proposed Dataset.

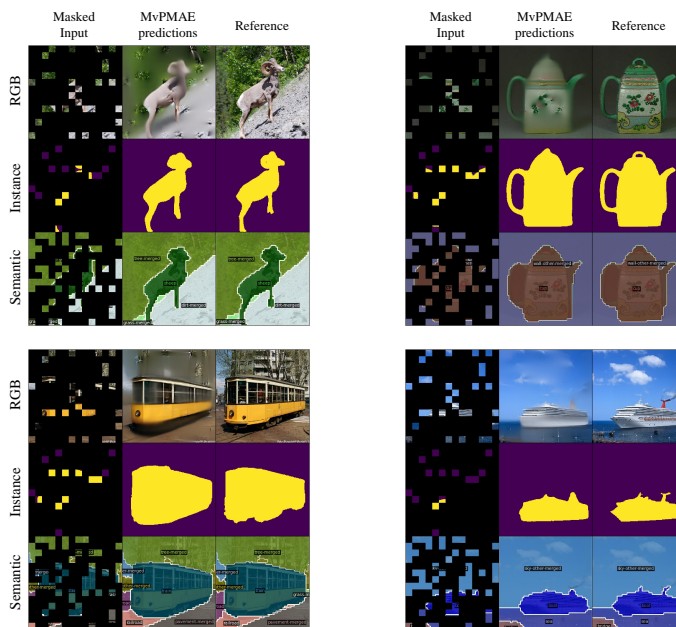

Figure 7: Illustration of reconstructions on randomly masked multi-modal data.

## D.3 SINGLE-MODAL PREDICTION WITH RGB IMAGES

For single-modality predictions, we start by utilizing fine-grained RGB images to predict coarse-grained instance masks and semantic masks. As illustrated in Figure 8, even when RGB images are the sole input, our model demonstrates remarkable accuracy in predicting both object and semantic information within the images. Impressively, it can even correct errors present in the reference pseudo-labels. This highlights the capability of our framework to help the model learn rich and robust semantic representations from the images.

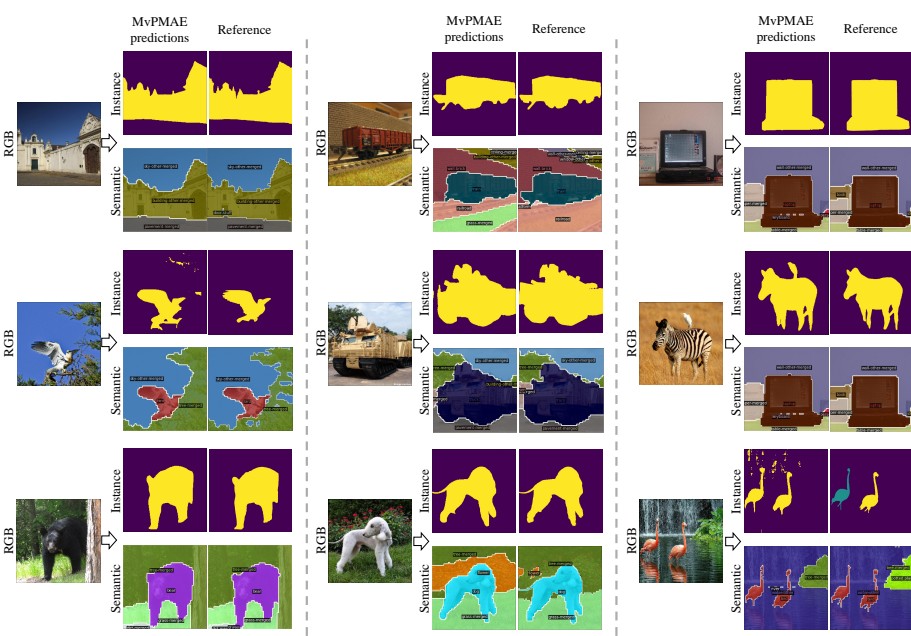

Figure 8: Illustration of single-modal prediction with RGB images.

## D.4 SINGLE-MODAL PREDICTION WITH INSTANCE MASKS

We further investigate the use of instance masks to predict information in other modalities. As shown in Figure 9, since instance masks primarily capture object contours and boundary details, the predicted RGB images tend to lack texture, resembling 3D models without texture rendering. On the other hand, when predicting from semantic masks, our model infers object categories based on their contours and imaginatively predicts surrounding scene details using knowledge learned from the multi-modal dataset. This highlights the model's strong capability to capture and utilize interactive information across diverse modalities.

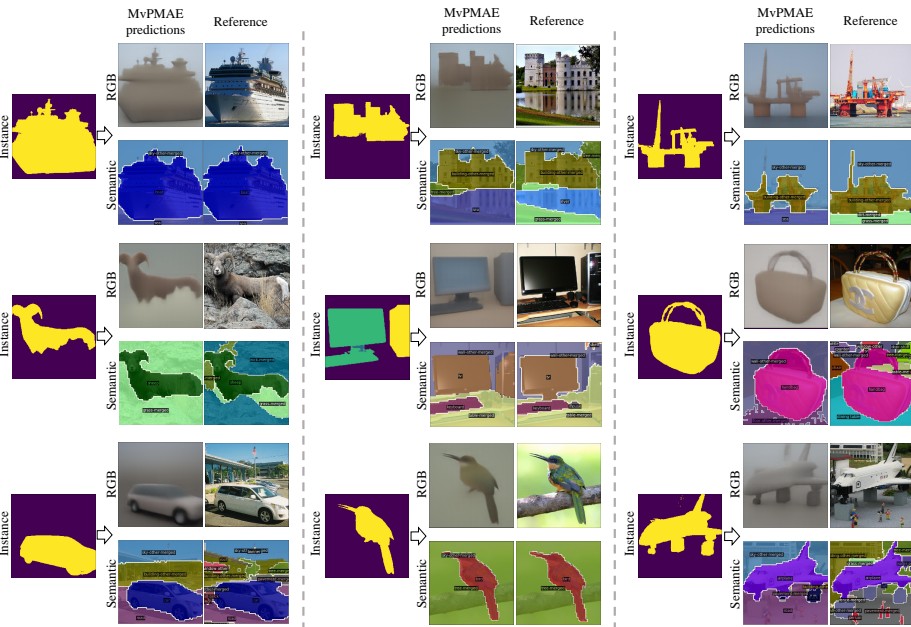

Figure 9: Illustration of single-modal prediction with instance masks.

## D.5 SINGLE-MODAL PREDICTION WITH SEMANTIC MASKS

When using semantic masks to predict instance masks, as shown in Figure 10, our model effectively captures the primary objects in the image and accurately predicts their contours. In the case of RGB image prediction, the incorporation of semantic information enables the generated RGB images to exhibit basic texture rendering corresponding to the semantics of different regions, compared to using instance masks alone. This highlights our model's inherent ability to decouple object contours, region boundaries, and semantic information.

# E DETAILED EXPERIMENT SETTINGS

## E.1 PRETRAINING

Table 7 shows the default configuration. Our framework employs Vision Transformer (Dosovitskiy et al., 2021) as the backbone network, processing $224 \times 224$ input images from ImageNet-1K (Russakovsky et al., 2015). We utilize the ViT-B model with patch size of 16, adopting a masking ratio of 1/6. The input data include three modalities: RGB images, instance masks, and semantic masks. We design a dedicated decoder comprising one cross-attention and two self-attention transformer blocks, featuring a dimensionality of 256 with 8 attention heads. The models are trained for 400 or 1600 epochs, including a 40-epoch warmup phase, with a total batch size of 2048. We employ the AdamW (Loshchilov & Hutter, 2019) optimizer with a base learning rate of $1 \times 10^{-4}$, linearly scaled as lr = base_lr × batch_size/256. The optimization parameters include a weight decay of 0.05, momentum parameters $\beta_1 = 0.9$ and $\beta_2 = 0.95$, and a cosine learning rate decay schedule. Standard

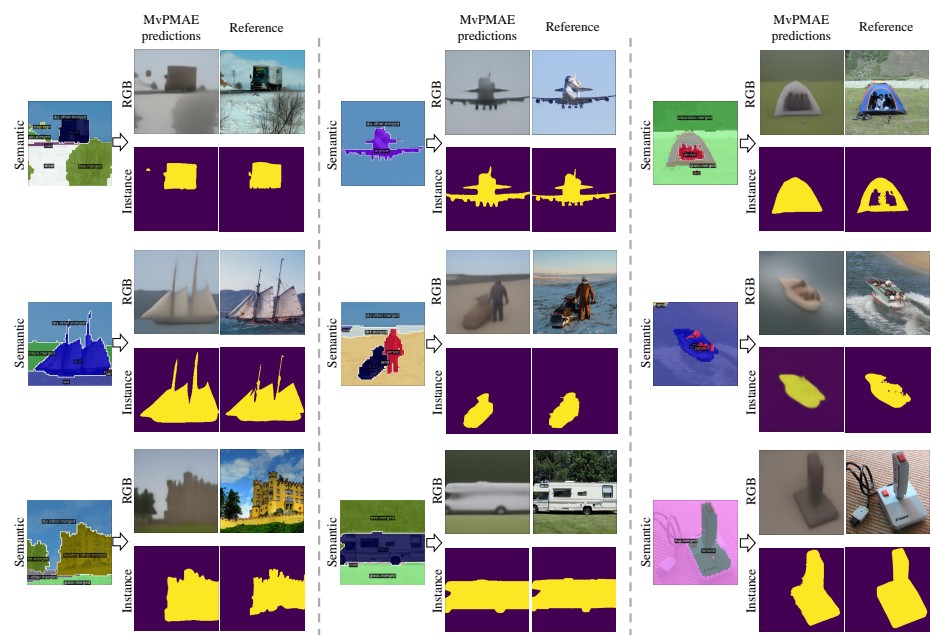

Figure 10: Illustration of single-modal prediction with semantic masks.

data augmentations such as random cropping and horizontal flipping are applied. All Transformer blocks are initialized using Xavier uniform initialization (Glorot & Bengio, 2010), following the MAE (He et al., 2022) approach.

Table 7: Pre-training settings.

| config | ViT-B |
|---|---|
| optimizer | AdamW |
| base learning rate | 1e-4 |
| weight decay | 0.05 |
| optimizer momentum | $\beta_1, \beta_2 = 0.9, 0.95$ |
| batch size | 2048 |
| learning rate schedule | cosine decay |
| pre-training epochs | 400/1600 |
| warmup epochs | 10/30, 10/40; 10/40 |
| augmentation | random cropping& horizontal flip |
| mask ratio | 1/6 |
| pre-training resolution | $224 \times 224$ |

### E.2 IMAGE CLASSIFICATION

The default configuration is shown in Table 8. We conduct end-to-end supervised fine-tuning on the ImageNet-1K dataset at $224 \times 224$ resolution, adhering to standard practices for fair method comparison. For ViT-B, we train for 100 epochs with 5 warmup epochs, employing base learning rates of $1e-3/5e-4$ and layer-wise learning rate decay of 0.7/0.65 for 400/1600 epochs, respectively. The training configuration maintains a batch size of 2048 and a drop path rate of 0.1 (Huang et al., 2016). Robust data augmentation techniques are applied, including label smoothing (Szegedy et al., 2016), mixup (Zhang et al., 2017), cutmix (Yun et al., 2019), and randAugment (Cubuk et al., 2020). Following MAE, we replace class tokens with global pooling features during fine-tuning. The learning rate adheres to the linear scaling rule: lr = base_lr × batch_size/256.

Table 8: Fine-tuning settings for image classification.

| config | ViT-B |
|---|---|
| optimizer | AdamW |
| base learning rate | 1e-3(400e);5e-4(1600e) |
| weight decay | 0.05 |
| optimizer momentum | $\beta_1, \beta_2 = 0.9, 0.999$ |
| layer-wise decay | 0.7(400e);0.65(1600e) |
| batch size | 2048 |
| learning rate schedule | cosine decay |
| training epochs | 100 |
| warmup epochs | 5 |
| augmentation | RandAug (9, 0.5) |
| label smoothing | 0.1 |
| mixup | 0.8 |
| cutmix | 1.0 |
| drop path rate | 0.1 |
| fine-tuning resolution | $224 \times 224$ |

### E.3 OBJECT DETECTION AND INSTANCE SEGMENTATION

Table 9 illustrates The default setup. We integrate the pre-trained ViT backbone into the Mask R-CNN (He et al., 2017) framework, conducting fine-tuning on the COCO (Lin et al., 2014) dataset using the MMDetection (Chen et al., 2019) implementation. The adaptation involves multi-scale training, resizing images to have a short side between 480 and 800 and a long side no greater than 1333. We employ the AdamW optimizer with a learning rate of $3e-3$, weight decay of 0.05, and total batch size of 16. Layer-wise decay rates are 0.75, with drop path rates of 0.2, respectively. We utilize a $1\times$ training schedule of 12 epochs, decaying the learning rate by $10\times$ at epochs 9 and 11. Performance is evaluated on COCO val2017 using bounding box $AP^b$ and mask $AP^m$ metrics.

Table 9: Fine-tuning settings for object detection and instance segmentation.

| config | ViT-B |
|---|---|
| optimizer | AdamW |
| base learning rate | 3e-3 |
| weight decay | 0.05 |
| optimizer momentum | $\beta_1, \beta_2 = 0.9, 0.999$ |
| layer-wise decay | 0.75 |
| batch size | 16 |
| learning rate schedule | step decay |
| training epochs | 12 |
| drop path | 0.2 |

### E.4 SEMANTIC SEGMENTATION

The default setup is depicted in Table 10. We incorporate the pre-trained ViT-B backbone into the UperNet (Xiao et al., 2018) architecture for semantic segmentation on the ADE20K (Zhou et al., 2017) dataset. The fine-tuning process spans 160k iterations with $512 \times 512$ input resolution, utilizing the AdamW optimizer. Key training parameters include a base learning rate of $4e-4$, weight decay of 0.05, and batch size of 16. The learning rate follows a warmup of 1500 iterations before linear decay. Segmentation performance is evaluated using mIoU on the validation set.

Table 10: Fine-tuning settings for semantic segmentation.

| config | ViT-B |
|---|---|
| optimizer | AdamW |
| base learning rate | 4e-4 |
| weight decay | 0.05 |
| optimizer momentum | $\beta_1, \beta_2 = 0.9, 0.999$ |
| layer-wise decay | 0.65 |
| batch size | 16 |
| learning rate schedule | linear decay |
| training iterations | $160k$ |
| drop path | 0.1 |

