# OpenReview forum: "CFMAE: A Coarse-to-Fine Vision Pre-training Framework for Hierarchical Representation Learning"
_ICLR.cc/2026/Conference — ICLR 2026 Conference Withdrawn Submission_

### Official Review · Reviewer_4119 · 2025-10-17

**Soundness:** 2
**Presentation:** 2
**Contribution:** 2
**Rating:** 4
**Confidence:** 3

**Summary:**

This paper introduces CFMAE, a vision pre-training framework designed to unify the strengths of contrastive learning (CL) and masked image modeling (MIM). CFMAE leverages a coarse-to-fine strategy across three visual granularities—semantic masks (coarse), instance masks (intermediate), and RGB images (fine)—to learn hierarchical representations. The framework features a cascaded decoder that sequentially reconstructs scene-level semantics, object-level instances, and pixel-level details, guided by a progressive masking strategy that transitions from semantic to instance to random masking. Extensive experiments on ImageNet-1K, COCO, and ADE20K demonstrate CFMAE’s superior performance in image classification, object detection, and semantic segmentation, as well as improved robustness on out-of-distribution ImageNet benchmarks.

**Strengths:**

* The paper provides thorough experiments across multiple tasks and datasets, showing consistent improvements over strong baselines like MAE and MultiMAE.

* The paper includes detailed component-wise analysis, validating the contribution of each architectural and training designs.

**Weaknesses:**

* **Inferior performance and increased complexity than existing feature-distillation-based MIM approaches**. The proposed approach (CFMAE) requires an external model (SAM, as mentioned in Appendix B) to generate semantic segmentation and instance segmentation masks. This makes it more like a distillation method. However,
   -  Some existing distillation-based MIM pretraining approaches such as MILAN [1] and MaskAlign [2] are missing in comparison, and CFMAE has an inferior peformance than them.
   - What are the strengths of CFMAE in comparison with simply distilling the features from SAM?  The latter is simpler and could even perform better according to the experience from MILAN and MaskAlign.  CFMAE is possibly overdesigned.

* **Sensitivity to hyperparameters and masking schedule**. CFMAE introduces several hyperparameters (such as $\alpha_I$ and $\alpha_S$ in Eq. 4, and $\lambda_S$, $\lambda_I$, $\lambda_R$ in Eq. 5) and a mannually determined masking schedule (Section 3.3 and Appendix C), which may require careful tuning. Sensitivity to the masking schedule is not discussed.

[1] MILAN: Masked Image Pretraining on Language Assisted Representation, https://arxiv.org/pdf/2208.06049.
[2] Stare at What You See: Masked Image Modeling without Reconstruction. CVPR 2023.

**Questions:**

Different from RGB input, semantic segmentation masks and instance segmentation masks are single-channel. How do you map them to $D$-dimensional embeddings? This technical detail is missing in Section 3.1.

---

### Official Review · Reviewer_9Qxe · 2025-10-25

**Soundness:** 2
**Presentation:** 2
**Contribution:** 2
**Rating:** 2
**Confidence:** 4

**Summary:**

This paper introduces a vision pre-training framework that combines a coarse-to-fine approach using a cascaded decoder and progressive masking strategy. The method aims to unify different self-supervised learning (SSL) approaches, including contrastive learning and masked image modeling, by leveraging multi-granular inputs (RGB images, instance masks, and semantic labels) to learn hierarchical representations. The proposed framework demonstrates strong performance on various tasks, including ImageNet classification, COCO detection, and ADE20K segmentation.

**Strengths:**

- The coarse-to-fine design with a cascaded decoder effectively enforces structured feature refinement, outperforming parallel decoders like MultiMAE.
- The progressive masking strategy dynamically shifts focus from semantics to details, enhancing hierarchical learning.
- The framework exhibits robustness and generalizability across multiple tasks, with notable efficiency gains.

**Weaknesses:**

The main concern with this paper is that it claims to be an SSL method, but uses a powerful SAM model to annotate the pre-training data. This raises two issues:

1. The paper compares itself to SSL methods that do not use labeled data or only use image-level labels. This comparison is unfair because the proposed method relies on a strong SAM model for pre-training.
2. The use of SAM for pre-training can be seen as a form of hard distillation, rather than a true SSL algorithm. This is because SAM is known for its strong performance on dense prediction tasks with large-scale pre-training datasets.

I am astonished that the authors placed such crucial information in the appendix.

The insights on CL and MIM lack detailed information. Moreover, the idea of combining both methods is not novel, as CMAE [1] has already integrated them. Unfortunately, the paper does not provide any discussion or comparison with CMAE.

The analysis in Fig. 1 bears resemblance to attention artifacts observed in prior work [2], but the authors do not provide a thorough analysis or comparison. Meanwhile, there are numerous studies analyzing attention artifacts during pre-training. In this regard, the advantages of this paper remain unclear.

The paper omits discussions and comparisons with related SSL papers, such as Fast-iTPN [3] and MAP [4].

Can this method be generalized to other architectures, such as ConvNets and Mamba?


Refs:
[1] Contrastive masked autoencoders are stronger vision learners. TPAMI 2023.

[2] Vision Transformers Need Registers. ICLR 2024.

[3] Fast-iTPN: Integrally pre-trained transformer pyramid network with token migration. TPAMI 2024.

[4] MAP: Unleashing Hybrid Mamba-Transformer Vision Backbone's Potential with Masked Autoregressive Pretraining. CVPR 2025.

**Questions:**

Please see the main weaknesses.

---

### Official Review · Reviewer_Vyhm · 2025-10-28

**Soundness:** 3
**Presentation:** 2
**Contribution:** 2
**Rating:** 4
**Confidence:** 5

**Summary:**

The paper introduces CFMAE, a masked autoencoding framework that builds visual representations coarse→fine by pairing a cascaded decoder (predicting semantic masks, then instance masks, then RGB) with a progressive masking schedule that transitions from semantic-guided to instance-guided to random as training proceeds, all atop a shared ViT encoder. It supports this design by constructing a multi-granular ImageNet-1K derivative: for each of the 1.28M images, instance masks are produced via a Grounded-SAM–style two-stage pipeline (GroundingDINO→SAM/HQ-SAM) and semantic masks via SEEM, organized under a 133-class (80 thing / 53 stuff) ontology. The model is trained with a multi-task objective—cross-entropy for semantic and instance predictions plus MSE for RGB reconstruction.

**Strengths:**

* The paper advances masked image modeling with a coarse→fine formulation that is conceptually tidy and practically motivated. Its two central ideas—a cascaded decoder that predicts semantic → instance → RGB and a progressive masking curriculum aligned to that order—are a coherent twist on multi-task MIM that encourages hierarchical feature formation (not merely parallel prediction).

* On clarity, the paper provides a concrete training/evaluation recipe and a readable ablation table that traces how each component (dataset, decoder, masking) contributes to the final accuracy, making the takeaways easy to audit.

**Weaknesses:**

* **Weak baselines:** In Tables 1,2, and 3, there are many baselines, all for before 2025, and only one baseline for 2024. There are more new baselines that have better accuracy, which is why the author did not compare them with them like in the FOLK paper [1]. I think if the author compares with newer baselines, their model can't outperform or at least achieve similar results with more computational cost. There is a doubt about the model's performance compared with the new baselines!


* **Comparability concerns when using multi-granular inputs at fine-tuning:** Some of the big results (e.g., 84.4% top-1) use extra semantic/instance inputs during fine-tuning, not just RGB. Many prior methods fine-tune with RGB only, so it’s not an apples-to-apples comparison.


* **Novelty vs. prior multi-task/multi-modal MIM is under-differentiated:** The method combines a cascaded multi-task decoder with a guided→random masking curriculum, but the paper does not rigorously isolate what is fundamentally new relative to earlier multi-modal/semantic-guided MIM lines (e.g., parallel multi-head decoders, deep supervision, semantic-aware masking). The main evidence is that CFMAE > MultiMAE and MAE in reproduced runs, but this can still reflect implementation/recipe differences rather than a conceptual step.


* **Grounding/attention evidence is mostly qualitative:** Claims about “hierarchical representations” and resolving “attention drift” rely on visualizations (Figure 1) rather than quantitative localization. Actionable: add mask-token IoU between predicted attention maps and ground-truth pseudo-masks, or phrase↔region alignment probes, plus failure galleries illustrating where the cascade helps vs. hurts







[1] Monsefi, Amin Karimi, et al. "Frequency-Guided Masking for Enhanced Vision Self-Supervised Learning." The Thirteenth International Conference on Learning Representations (ICLR 2025).

**Questions:**

* Novelty and positioning: What is the precise conceptual difference between your cascaded decoder and prior multi-task/multi-modal MIM (e.g., parallel decoders with deep supervision or semantic-guided masking)? Please provide a controlled comparison where capacity, tokens, and training budget are strictly matched, isolating “cascade vs. parallel” as the only change.


* Practicality without extra labels: Many users won’t have semantic/instance masks for fine-tuning. Can CFMAE’s benefits be realized when downstream training is strictly RGB-only? If yes, summarize those results; if not, discuss implications and recommended usage.


* Failure modes: Please include a short failure gallery with class/sparsity breakdowns (e.g., small/thin objects, heavy clutter, overlapping instances), and diagnose whether errors arise in the semantic stage, the instance stage, or RGB reconstruction.

* Masking curriculum specifics: Please detail the exact schedules for semantic-guided → instance-guided → random masking (phase lengths, masking ratios, αS/αI trajectories). How sensitive are results to these schedules? A small grid or curves would help.


**I am open to changing my score based on the author's responses.**

---

### Official Review · Reviewer_sHDF · 2025-10-31

**Soundness:** 2
**Presentation:** 2
**Contribution:** 2
**Rating:** 4
**Confidence:** 4

**Summary:**

The paper proposes CFMAE, a hierarchical self-supervised pre-training method that combines masked image modeling (MIM) and contrastive learning (CL) ideas under a coarse-to-fine scheme. CFMAE reconstructs semantic masks, instance masks, and RGB images using a cascaded decoder and a progressive masking schedule. Pseudo-labels for semantics and instances are generated on ImageNet using SAM, Grounded DINO, and HQ-SAM. The authors claim this unifies coarse and fine-grained cues, mitigating the “attention drift” seen in MAE-style methods.

**Strengths:**

While the conceptual originality is limited, the paper is at least well-written and the hierarchical pre-training formulation (semantic → instance → RGB) is clearly presented. The integration of a cascaded decoder and a progressive masking schedule follows a logical “coarse-to-fine” rationale, and the paper includes a full ablation section that is easy to follow.

**Weaknesses:**

- Methodological setting raises fundamental concerns.
  - The entire training relies on pseudo-labels generated by SAM, Grounded DINO, and HQ-SAM, which introduces strong external supervision and heavy computational cost. If the pre-training or inference pipeline depends on these models, the method is no longer self-supervised in practice.
  - If such large-scale pretrained models are already available, it is unclear why one would retrain a ViT on ImageNet using their outputs. The paper should justify what new information CFMAE learns beyond what those models already encode.
  - One motivation for unsupervised pre-training is scalability to Internet-scale data. However, running expensive segmentation models to generate multi-granular masks removes this scalability advantage. The claimed efficiency is therefore questionable.
  - **Constructive suggestion**: One way to demonstrate practical value is to train CFMAE using only a small subset of pseudo-labels (e.g., 1 %, 5 %, 10 % of ImageNet) and show that even limited hierarchical guidance improves performance. That would strengthen the empirical message that “a small amount of structured information helps.”

- Outdated problem framing (CL vs MIM debate).
  - The motivation is positioned around resolving limitations of contrastive learning versus masked image modeling, which feels outdated by late 2025.
  - The paper does not acknowledge recent unified or self-distillation approaches (e.g., DINO v2, iBOT v2, I-JEPA, data2vec 2.0), which already achieve strong global–local alignment without explicit hierarchical reconstruction.
  - TODO: A methodological comparison with DINO v2 on a consistent ImageNet setup would be valuable, both conceptually and empirically.

-  Limited validation of claimed “semantic-aware correspondence.”
    - The paper states that CFMAE yields better semantic-aware correspondences, which is plausible, but it provides no evaluation on tasks where such property matters, e.g., Semantic matching or part-level correspondence (SPair-71k, PF-Pascal) or Video object segmentation or tracking benchmarks. Without these, the claimed representational improvement remains speculative.
    - TODO: add those experiments

- Modest ImageNet gains
  - (ViT-B, +0.4–0.8% vs strong baselines) relative to the added pipeline complexity; larger relative wins appear mainly on dense tasks, but the paper does not isolate how much comes from the new dataset vs architecture/masking (Table 5 only partially decomposes this).

**Questions:**

N/A

---

### Note · Authors · 2025-11-20

I have read and agree with the venue's withdrawal policy on behalf of myself and my co-authors.